

# Storm-wave trends in Mexican waters of the Gulf of Mexico and Caribbean Sea

Elena Ojeda[1,3], Christian Mario Appendini[2,3], Ernesto Tonatiuh Mendoza[2,3]

[1] CONACYT - Instituto de Ingeniería, Universidad Nacional Autónoma de México, Puerto de Abrigo s/n, 92718, Sisal, México
[2] Instituto de Ingeniería, Universidad Nacional Autónoma de México, Puerto de Abrigo s/n, 92718, Sisal, México
[3] Laboratorio Nacional de Resiliencia Costera, Laboratorios Nacionales CONACYT, México

*Correspondence to*: Elena Ojeda (eojedac@iingen.unam.mx)

**Abstract.** Thirty-year time series of hindcast wave data were analysed for ten coastal locations along the eastern Mexican coast to obtain information about storm events occurring in the region, with the goal to examine the possible presence of interannual trends in the number of wave storm events and their main features (wave height, duration and energy content). The storms were defined according to their significant wave height and duration, and the events were classified as related to either tropical cyclones or Norte events. The occurrence and characteristics of both types of events were analysed independently. There is no statistically significant change in the number of wave storm events related to Nortes or their characteristics during the study period. However, there is a subtle increase in the number of events related to tropical cyclones in the western Caribbean region and a more evident increase in wave height and energy content of these events.

## 1 Introduction

Over the past decades, there has been increasing awareness of the effects of climate change in coastal regions, with numerous studies focused on possible implications of sea level rise (Nicholls, 2002; Wong et al., 2014) and on the modification of the intensity, frequency and location of storms worldwide (Kossin et al., 2014), which represent a more immediate consequence of climate change. Albeit these numerous studies, there is still low confidence in the results of large-scale trends in storminess over the last century (IPCC, 2013), mainly due to changes in the capabilities of observing techniques, which confound the possible presence of trends. One region with reliable data is the North Atlantic, where a robust increase in the frequency and intensity of the strongest storms has been observed since the 1970s, although there is still debate over the cause of this increase (IPCC, 2013; Webster et al., 2005).

Contrary to the meteorological effects of storms, where the main interest is focused on the location of the storm strike (e.g. Elsner et al., 1999; Simpson and Lawrence, 1971), wave conditions related to a storm event can be observed along distant coasts, well beyond the region of wind stress. Hence, a modification in storm characteristics will be associated with inherent changes in wave height and the storm surge reaching the coastline, which are the two main factors responsible for considerable economic losses in coastal and offshore areas (Mendelsohn et al., 2012; Neumann et al., 2014). This makes the investigation of changes in storminess trends a mayor concern for coastal management, even more so when taking into consideration that possible storm effects might be enhanced by the effect of sea-level rise and the increase in coastal development.

Wave conditions are commonly obtained from in situ observations (buoys, tide gauges, ships), satellite altimeter data or wave models. In the U.S. Gulf of Mexico (GoM) region, Komar and Allan (2008) analysed a 28-year register of measured buoy data in the central region of the Gulf of Mexico, but did not find long-term changes in wave height during the summer months, which, according to the authors, would be attributable to an increase in wave heights caused by tropical cyclones (TCs). This dataset was also included in a study by Bromirski and Kossin (2008) which extended the analysed data to three deep-water (>1000 m) buoy registers in the Northern GoM, each covering a period of 28 years. Their study demonstrated the relationship of such long-term registers with shorter-term coastal registers, making their findings applicable to near-coastal regions. In their



study, TC waves were separated from the rest of the register using the National Hurricane Center best-track record and an increase in the number of TC-related events over the last decades was found. However, this increase was neither associated with increases in significant wave height related to the TC events, nor with the duration of the TC events. Their study also found a shift in the intra-annual distribution of TC-related events between the first and second parts of the register, with the

majority of events taking place during August and September in the first half of the register and during September and October in the second half of the register.

Along the coast of the Mexican GoM, extreme wave storm events are mainly caused by tropical cyclones and Nortes. Nortes are anticyclonic cold surges that enter the Gulf of Mexico from North America, generating strong northern winds and, therefore, present ideal conditions for fetch causing mature wind waves. The study of the occurrence and interannual trends of

extreme wave storm events related to both TC and Nortes must be regarded separately, given that possible long-term changes in the behaviour of TC and Nortes and their responses to climate change are not expected to be analogous (Komar and Allan, 2008).

As for the western Caribbean Sea (WCS), storminess trend studies based on wave datasets are absent to the authors' knowledge. Projected changes in wind shear suggest a decrease in the number and intensity of TCs in the region (Biasutti et al., 2012),

although studies suggest an intensification of wind speed of higher-category events on a global scale (Holland and Bruyère, 2014). Wave condition models do not project increases in wave height in this area (Appendini et al., 2014), although no analyses specifically for extreme events exist.

Along the eastern coast of Mexico (Gulf of Mexico and Western Caribbean Sea), in situ and satellite data are scarce and temporally discontinuous. This contribution analyses a 30-year time series of hindcast wave data, covering the period of 1979

to 2008 (Appendini et al., 2013). The selection of this time interval allows for comparison to previous work in the region and is considered the standard period by the World Meteorological Organization to characterize climate, although the use of shorter time periods has been suggested to characterize non-stationary climates, i.e. under climate change conditions (Arguez and Vose, 2011). Appendini et al. (2013) did not find any significant trend in time series of extreme wave heights (using the 99th percentile) along the eastern Mexican coast, but their research examined the entire time series without regard of the type of

extreme event.

The aim of this work is to determine possible interannual trends in the number of wave storm events and their main features (wave height, duration and energy content). This was achieved by first identifying extreme wave storm events for a number of near-coastal locations along the eastern Mexican coast, and then classifying them according to the meteorological conditions that caused these events (TC or Nortes).

## 2 Study area

The eastern coast of Mexico extends along the GoM and the WCS. The study area is within the region with the highest correlation coefficient (best performance) of the hindcast data, and comprises eight nodes located in the Mexican GoM at 50-m depth, and two nodes in the WCS within 100-m depth. Additionally, the nodes were selected based on the proximity of coastal settlements: Matamoros, Tampico, Veracruz, Coatzacoalcos, Paraiso, Campeche, Progreso, Holbox, Cancun and

Tulum (Fig. 1).

Campeche, Puerto Progreso and Holbox are characterized by an ubiquitous wide continental shelf of approximately 245 km with a slope of 1/1000 and a low-lying topography of the adjacent emerged areas (Enriquez et al., 2010).

Wave conditions are generally mild with mean *significant wave heights* (SWH) around 1 m, although wind waves with maximum individual wave heights of approximately 27 m have been measured by wave gauges in the northern GoM (Wang

et al., 2005) during Hurricane Ivan. The main meteorological systems contributing to storm wave conditions are mid-latitude anticyclonic meteorological systems that generate northerly cold fronts known as Nortes with an average of 11 to 21 Nortes



per season (Reding, 1992), and cyclonic systems that include tropical depressions, tropical storms and hurricanes. There is some overlap of the occurrence of either type of event during the year and the two systems can interact, mainly in October. The North Atlantic tropical cyclone season runs from 1 June to 30 November, with September being the month with the highest number of tropical cyclones directly affecting the Mexican coasts (Rosengaus and Vázquez, 2002). Norte events usually occur

from October to April, with the most intense events taking place from December to March (Appendini et al., 2014).

### 3 Datasets

Three datasets are used in this study: 1) 30 years of hindcast wave data, 2) the International Best Track Archive for Climate Stewardship (IBtracs) dataset for discerning TCs affecting the study area, and 3) a Norte dataset designed for the GoM and WCS regions.

### 3.1 Hindcast wave data

The hindcast wave data cover the period from 1979 to 2009 in three-hour intervals for the entire eastern Mexican coast (Appendini et al., 2013). We used the 30-year wave data in order to comply with the World Meteorological Organization's recommendation for characterizing non-stationary climates, i.e. under climate change conditions (Arguez and Vose, 2011). As mentioned before, eight nodes located in the Mexican GoM at 50-m depth and two nodes in the WCS at 100-m depth were

evaluated. These isobaths were selected to assure that the nodes are in deep water to avoid any shoaling, refraction and bottom friction.

### 3.2 IBtracs

The IBtracs (Knapp et al., 2010) record was used given that this platform includes the most complete set of historical TCs available. The data were downloaded from the NCDC website (https://www.ncdc.noaa.gov/ibtracs) in May 2016.

### 3.3 Norte dataset

The Norte dataset includes the dates when Nortes entered and left the Gulf of Mexico. This dataset was derived from the Climate Forecast System Reanalysis (CFSR) data (Saha et al., 2010), and the Nortes are identified based on the pressure difference between Yucatan and Texas, as well as wind speed at different positions over the Gulf of Mexico.

### 4 Methodology

In this study, the occurrence and interannual trends of extreme wave storm events caused by TCs and Nortes were regarded separately, given that their behaviour and response to climate change are not expected to be analogous (Komar and Allan, 2008). The present study of the GoM and WCS wave data examines the use of a common tool for coastal management where the dataset is inspected looking for individual storm events based on the register of significant wave heights, establishing a minimal duration for storm events (Mendoza et al., 2011) and associated meteorological conditions based on the identification

and separation of TC and Nortes events. This approach not only provides improved insight into wave formation compared to using a summer and winter events distribution (Komar and Allan, 2008), but also allows further analyses of the number, energy content and duration of storm events, as well as possible trends associated with a given type of meteorological storm.

### 4.1 Storm definition

From an oceanographic point of view, a storm can be defined as an increase in wave height and sea level (storm surge)

exceeding a certain threshold during a certain amount of time (e.g. Mendoza et al., 2011). A common way to characterize





extreme events is using a *Peak-Over-Threshold* methodology, which implies that an extreme wave event (storm event) occurs when the significant wave height exceeds a given threshold during, at least, a certain period of time.

Conventionally, the selection of the wave height threshold takes local characteristics of the wave regime into account. In this study, a storm is defined as an event reaching a SWH greater than $H_{threshold}$ for at least 12 hours. $H_{threshold}$ is calculated for each

node as

$$H_{threshold} = \langle SWH \rangle + 2\sigma \qquad (1)$$

where the chevrons refer to average values and $\sigma$ is the standard deviation of the SWH time series. The SWH criterion is based on the methodology used by Walker and Basco (2011), while the minimum duration criterion is applied taking the reported decrease in the duration of Nortes events as a result of climate change (Pérez et al., 2014) into account. Finally, in order to

separate consecutive storm events and make certain that the events are statistically independent, a minimum time between consecutive events was established (48 hours). If the SWH is below the threshold for less than 48 hours, two consecutive events will be considered as one event.

For each of the identified storms, the following main characteristics were obtained: mean and maximum SWH, storm duration ($t_s$), mean storm energy content and maximum energy content. The $E_S$ is related to the storm beach erosion potential (Mendoza

and Jiménez, 2006), and is given by

$$E_s = \int_{t_1}^{t_2} SWH^2 \, dt \qquad (2)$$

where $t_1$ and $t_2$ define the storm duration and SWH > $H_{threshold}$. This equation leads to a more accurate value for $E_S$ than the traditional wave power equation that uses a single wave height value (usually the maximum SWH) to characterize the entire event.

## 4.2 Storm classification

In order to separate TCs from Nortes events, the IBtracs database was used to identify TCs, while Nortes were identified using the Norte identification index. Regarding TCs, two regions of influence were defined: the WCS region (62° to 89.5°W, 7° to 23°N), covering events affecting the Tulum and Cancun nodes, and the GoM region (81° to 100°W, 17° to 32°N) covering the rest of the nodes. Wave conditions are considered to be caused by a TC when an event occurred during the pass of a TC through

its regions of influence or during the following 36 hours. Regarding the Nortes, wave conditions are considered to be related to a Norte event when the initial storm event (wave conditions) started during the occurrence of the Norte.

For a considerable number of events (between 7 and 25, depending on the node), a wave storm event occurred during both types of events. To discriminate the responsible event in these cases, the first approach was to compare the number of hours during which the meteorological and wave events coincided. If one of the meteorological events had a larger number of

coincident hours, it was chosen to be the responsible event. If the hours were equal, the second approach was to look at the date of occurrence; events occurring from November to April were considered as Nortes, while those occurring from July to September were considered as TCs. When the events occurred in the months of May, June and October, the wind direction related to the maximum SWH of the event was considered. Winds coming from N230°E to N45°E were considered caused by Nortes, while the rest of the events were considered TCs. Events that did not correspond to either type were not considered in

this study.

## 4.3 Decadal trends of event characteristics

Time series of the event characteristics per season evaluated for the TC and Nortes series are: the number of storm events, the mean and maximum SWH ($SWH_{mean}$ and $SWH_{max}$), the mean and maximum $E_s$, ($E_{s,mean}$ and $E_{s-max}$), the mean duration of the storm events and the sum of storm event durations.



A Mann-Kendall trend test (Kendall, 1975; Mann, 1945) was performed for the time series of the different evaluated characteristics of the storm events caused by TCs and Nortes. This non-parametric test is used to identify if there is a monotonic upward or downward trend through time, but it does not specify whether the detected trend is a linear or nonlinear trend. The null hypothesis assumes that the data are independent and identically distributed over time. Mann-Kendall tests were performed

in order to confirm the occurrence of trends that were significantly different from zero at the 90 % confidence level or higher. Following Casas-Prat and Sierra (2010) the $E_s$ and SWH data were log-transformed, which implies that the relationships obtained by linear regression between $E_s$ and SWH and time become exponential. In the case of the TC time series, a considerable number of years with no TC occurrence is present in the different datasets. For this reason, the probability of storm occurrence is introduced in the analysis to avoid biasing the data with the zero values. The SWH$_{mean}$, SWH$_{max}$, $E_{s,mean}$

and $E_{s,max}$ time series are analysed removing the zero data to obtain the temporal trends conditioned to storm occurrence:

$$\hat{E}_s(t|storm) = \ln \hat{a}_E t + \hat{b}_E \tag{3}$$

where $a_E$ and $b_E$ are the regression coefficients and ^ stands for predicted. The probability of storm occurrence ($\hat{p}_s(t)$) is estimated for the binary TC time series (1 = event occurred, 0 = no event occurred) by binomial logistic regression. The product of $E_s$ (t|storm) and $\hat{p}_s(t)$ results in the estimated temporal evolution of the $E_s$:

$$\hat{E}_s(t) = \hat{p}_s(t)\exp(\hat{a}_E t + \hat{b}_E) \tag{4}$$

In this contribution, the term "trend" is defined, for the number of storm events and their duration, as the slope of the linear relationship (LR) between each variable and time. For SWH$_{mean}$, SWH$_{max}$, $E_{s,mean}$ and $E_{s,max}$, calculated trends are complex and are simplified as the mean rate of annual increase, calculated as the slope between the initial (t = 1979) and final (t = 2008) estimations.

## 5 Results and discussion

### 5.1 Data overview

The largest SWH values in the region occur in Matamoros, with <SWH> of 1.34 m, clearly exceeding the 1.06 m occurring at the closest southern node, Tampico (Table 1). Mean monthly SWH during the study period show an analogous distribution for the GoM nodes, with lower values during the summer months reaching a minimum in August and higher values during the

winter months, from November to March (Fig. 2). The wave registers in the WCS nodes (Cancun and Tulum) present similar overall statistics (Table 1), but differ in the mean monthly SWH distribution (Fig. 2), as the yearly variability in SWH is not as apparent as in the GoM nodes and there is not a clear summer/winter pattern.

### 5.2 Storm classification

Storms are defined for each node based on its SWH time series. The critical SWH value (H$_{threshold}$) that defines storm events

varies from 1.87 m in Tulum to 2.56 m in Matamoros; for all cases, H$_{threshold}$ is exceeded for less than 5 % of the data. The total number of storms identified at the different nodes varies from 245 in Cancun to 407 in Matamoros (Fig. 3).

The classification into Nortes and TC-related events shows that between 5 % and 13 % of the events were caused by TCs. Nortes were responsible for between 51 % (Matamoros) and 88 % (Coatzacoalcos) of the events in the GoM. These numbers are significantly lower in the WCS, where Nortes were responsible for up to 28 % of the events (Fig. 3).

A number of events were not classified as related to Nortes or TC events and, therefore, were not considered in this study. This number varies greatly between nodes, from 5 % in the southern GoM to >58 % in the WCS (Fig. 3). An unexpected result is that a considerable number of events occurring in Matamoros are not related to any of the considered meteorological systems. This location is subject to the most restricted fetch among all analysed locations, so that storms are most likely related to other systems such as "suradas" or northerly winds that occur after the pass of a Norte. In Matamoros, suradas could occur while



the Norte is still in the Gulf of Mexico, generating southerly winds in most of the other locations. This requires more detailed analysis, particularly to establish the duration of Norte events at particular locations instead of in the GoM as a whole. The large percentage of unclassified events in the WCS is an expected result given that wave conditions in the WCS are largely influenced by the trade winds, which is supported by the findings of Appendini et al. (2014).

A more detailed view of the annual distribution of both types of events (Fig. 4) shows that GoM events resulting from the occurrence of Nortes are more abundant during January and December, while the number of events resulting from the occurrence of TCs is highest during September and October, although considerably smaller in number than the number of events caused by Nortes.

The WCS nodes differ in the number and monthly distribution of storm events. Although Tulum has a larger number of

10 registered events, fewer events are classified as caused by Nortes or TCs. This might be related to the location of Tulum, which is protected by Cozumel Island, and, due to the orientation of the coastline, is characterised by a small fetch for events approaching from all directions except easterly waves that make it prone to the occurrence of trade-wind-related events. Events resulting from Nortes in Cancun occur mostly from November to March, while in Tulum their number is highest from February to April. The distribution of events related to TCs is more similar, although the number of events is lower in Tulum.

Events caused by TCs start earlier in the WCS than in the GoM, with a few registered events in May and June. As already mentioned, in this region, the majority of the extreme wave events are not related to Nortes or TCs, and their occurrence is distributed throughout the year, with the largest number of unclassified events taking place in March (not shown).

The results of the annual distribution of TC- and Norte-related events (Fig. 4) demonstrate that the definition of the winter and summer season given by Komar and Allan (2008) is not applicable to the present study because a large percentage of events

would be excluded from the analysis. For example, during October both Norte- and TC-related storm events are common in the region.

The effect of TCs in the WCS is mostly observed during what is currently defined as the hurricane season, i.e. from June to the end of November, with a similar behaviour at the two easternmost nodes of the GoM (Progreso and Holbox). The northernmost nodes (Matamoros and Tampico) also indicate a hurricane-season-type distribution of events, with TC-related

events recorded during July and August, but not in November or December, showing a pattern more similar to the U.S. GoM results presented by Bromirski and Kossin (2008). However, the nodes in the Bay of Campeche (Veracruz, Coatzacoalcos, Paraiso and Campeche) show that events related to TCs occur between August and December, with no registered events during June or July and the majority of events taking place in September and October.

### 5.3 Decadal trends in the number of events and their characteristics

Nortes and TCs are evaluated by season: from May to December for TCs and from September to the following August for Nortes. For this reason, there are 30 seasons for TCs, but only 29 seasons for Nortes because the last season is only partially represented in the dataset.

As expected, the patterns of the number of events caused by Nortes and TCs differ (Fig. 5 and Fig. 6), as well as the trends and the Mann-Kendall test results obtained for each parameter, which are summarized in Table 2 and Table 3 for the Nortes

and TC events, respectively.

The time series of the number of storm events related to Nortes per season are given in Fig. 5. The trends obtained from a simple linear regression show a positive slope of less than 0.03 events $yr^{-1}$ in all cases and do not indicate a particular organization among the nodes in the study area (Table 2). Furthermore, according to the Mann-Kendall test results, none of the nodes shows significant trends (at the 90 % confidence level) in the number of events caused by Nortes.

There is more consistency in the time series of the number of events caused by TCs (Table 3). Although, according to the MK test results, significant positive trends in the number of storm events caused by TCs are only found at the Holbox, Cancun and



Tulum nodes, the entire study area shows a certain organization, with Paraiso and Coatzacoalco showing trends close to zero and increasing slopes of the regression lines with increasing distance from these nodes.

### 5.4 Events caused by Nortes

According to the Mann-Kendall test results at the 90 % confidence level, none of the nodes, with the exception of Matamoros, Tampico and Progreso, show significant trends for the intensity or the duration of the Nortes. The significant trends that were found show that: i) Matamoros shows a decrease in $SWH_{mean}$ and $SWH_{max}$, ii) Tampico shows significant trends for $E_{s,mean}$ and $E_{s,max}$ and iii) Progreso shows a decreasing trend in $SWH_{max}$.

Despite the Mann-Kendall test results, it can be observed that along the GoM region, $SWH_{mean}$ has not changed significantly, but $SWH_{max}$ trends (only confirmed by Mann-Kendall results in the Matamoros and Progreso nodes) show a decrease in maximum SWH with the smallest negative values of the trends (around 0.01 m yr$^{-1}$) found in Veracruz, Coatzacoalcos, Paraiso and Campeche and the largest negative values of the trends found in Matamoros and Progreso, with -0.03 m yr$^{-1}$ and -0.06 m yr$^{-1}$, respectively, which represents a decrease in $SWH_{max}$ of around 0.9 m and 1.65 m, respectively, during the study period. In the WCS, the trends in $SWH_{max}$ are overall positive, although not confirmed by the Mann-Kendall test.

The negative trends in $SWH_{max}$ in Matamoros and Tampico are accompanied by negative trends of $E_{s,mean}$ and $E_{s,max}$, with larger negative values in Matamoros, where there is also a negative trend for the duration of the events. However, the presence of these trends in energy content is not confirmed by the Mann-Kendall test. Only Tampico shows a confirmed trend in the $E_{s,mean}$ and $E_{s,max}$.

In general, the result of the analyses of energy content and duration of events caused by Nortes do not show significant trends for the majority of the nodes. There is no consistency at the regional scale (i.e. several proximal nodes showing similar results) nor in the presence of significant trends for a single node (e.g. Matamoros shows significant trends for the $SWH_{mean}$ and $SWH_{max}$ but those are not reflected in the $E_{s,mean}$ or $E_{s,max}$, which would be expected especially in cases where both SWH and duration are decreasing).

Previous research by Pérez et al., 2014 found a decrease in the duration and an increase in intensity of Norte events for future climate scenarios using the TL959L60-AGC model of the Meteorological Research Institute of Japan under the A1B scenario. A decrease in duration is observed in our results for the northernmost nodes of the GoM and in the WCS, but is not statistically significant based on the Mann-Kendall test performed on the results. However, this might be related to the definition of a storm event used in this work, which requires more than 12 hours with SWH $\geq$ H$_{threshold}$ to define an event, therefore biasing the effect of possible shorter events.

### 5.5 Events caused by TCs

The TC time series show a considerable number of years without the presence of storm events occurring at the different nodes. This number ranges between 43 % and 57 % of the analysed years, depending on the node (Fig. 6). This result is corroborated by Ramírez (1998), who investigated TCs arriving at the Yucatan Peninsula during the 1970–1995 period. In general, the percentage of years with no events related to TCs is larger during the first half of the study period, a circumstance more obvious for the WCS nodes, and in agreement with Bromirski and Kossin (2008), who found a general tendency for more significant wave events related to TCs since 1995, consistent with an increasing overall count of named storms during recent years. Klotzbach et al. (2015) attributed the decrease in the number of TCs from 1972 to 1992 and the following positive trend in the number of events to the Atlantic Multidecadal Oscillation.

The trends in the number of storm events shown in Fig. 6 reveals that, in addition to an increase in the probability of occurrence of a storm event ($\hat{p}_s(t)$) during the study period (Fig. 7), there is also an increase in the number of TC events per season. This, however, is not the case for the Coatzacoalcos and Paraiso nodes, which are the only two nodes where $\hat{p}_s(t)$ shows a negative slope (Fig. 7).





The estimated trends in the $SWH_{mean}$, $SWH_{max}$, $E_{s,mean}$ and $E_{s,max}$ time series of the events caused by TCs are influenced by: 1) the probability of occurrence of the events and 2) the trend conditioned to the occurrence of TC-related events [i.e. $\hat{E}_s(t|storm)$]. As already mentioned, $\hat{p}_s(t)$ shows positive slopes for all nodes with the exception of Coatzacoalcos and Paraiso (Fig. 7). The $\hat{E}_s(t|storm)$ trends (calculated as the slope between the first and last value of each time series) are given

in Table 4. The combination of both factors contributes to the trends in the time series of events caused by TCs (Table 3), which show, in general, larger values than the trends in events caused by Nortes (Table 2).

The effect of TCs in both regions is strongly influenced by the surrounding landmasses. In the GoM, the MK test only corroborates the presence of a trend in a low percentage of the time series (mainly those with analogous signals in the probability of occurrence and the trend related to the occurrence of events). Veracruz, Coatzacoalcos, Paraiso, Campeche and

10 Progreso do not show any trends according to the MK test. The Mann-Kendall test corroborates the presence of a trend in the $SWH_{mean}$ for the northern nodes (Matamoros and Tampico), with slopes of 0.05 m yr⁻¹ and 0.06 m yr⁻¹, respectively (in the case of Tampico, there is also a significant trend in the $SWH_{max}$ with a slope of 0.07 m yr⁻¹). However, the significance of the trends is not maintained for the storm energy contents, in which the effects of storm duration and the shape of the SWH register are considered. For example, in Matamoros, 2005 was the season with the largest number of events related to TCs, but the

15 duration of the single event that took place in 1995 was longer than the combined durations of the three events registered in 2005 or the two events registered in both 2002 and 2008. The event in 1985 corresponds to Hurricane Juan, a Category 1 hurricane that moved erratically in the GoM waters from 26 October to 1 November¨ causing $SWH_{max}$ at the GoM nodes of less than 5 m, but storm conditions with a duration of around five days (Fig. 8).

In contrast, the WCS nodes show significant trends according to the Mann-Kendal test results for all analysed parameters and

20 the effect of the increase in the number of TC events since 1995 is more obvious than for the GoM nodes (Fig. 9).

## 6 Summary

Based on the intra-annual distribution of storm events and the trends found for TC- and Norte-related events, the studied nodes can be separated into four regions:

i) the northernmost nodes (Matamoros and Tampico) are characterized by a TC season starting earlier than at the other nodes,

with a majority of TC events taking place in August and September and no registered TC events in November and December. The nodes show a similar monthly distribution of the occurrence of Norte- and TC-related events, similar values regarding the probability of TC occurrence with time and a certain consistency in the trends of TC-related events. However, there are also significant differences between these two nodes, the most noticeable being the large number of unclassified events occurring in Matamoros.

ii) the Bay of Campeche nodes (Veracruz, Coatzacoalcos, Paraiso and Campeche) show no events (of any type) occurring during June and July during the study period. The majority of events occur during the months of January, February and December, mostly associated with Nortes. Most of the TC-related events take place during September and October, but events have been recorded from August to December (with only one recorded event in December which corresponds to Tropical Storm Olga). These nodes do not show significant trends for any of the evaluated parameters, neither for Nortes nor for TCs.

iii) the Progreso and Holbox nodes show a behaviour partly comparable to the WCS nodes, mainly Holbox, which is closer to the WCS. In general, they record a lower number of storm events than the rest of the GoM nodes, with a slightly lower number of Norte-related events and a slightly larger number of TC-related events than the rest of the GoM region. The trends found for events caused by TCs present steeper slopes than (ii) but the Mann-Kendall test does not support the presence of a statistically significant trend.

iv) the WCS nodes (Cancun and Tulum), where the majority of events cannot be attributed to TCs or Nortes. In this region, the number of events related to Nortes is considerably lower and the majority of events related to TCs that have been recorded



occurred between July and October. Trends found for the Norte events are not significant according to the Mann-Kendall test, but trends for events related to TCs show an increase in number, duration and intensity during the study period.

## 7 Conclusions

In the GoM region, Nortes are responsible for the majority of extreme wave events occurring from 1 November to 30 April, while TCs are responsible for the majority of extreme wave events occurring during August. During the months of September and October, both Nortes and TCs can be responsible for extreme wave events. The TC season in the Mexican GoM starts and ends later than in the North Atlantic; it lasts from August to December instead of June to October.

There is not a general statistically significant change in the number of wave storm events related to Nortes or their characteristics in the Gulf of Mexico and the WCS during the study period. The time series of events related to Nortes and their characteristics do not show a consistent behaviour in the study area. Although there are a few time series where the presence of a trend in the data is corroborated by the Mann-Kendall test at the 90 % significance level, the available evidence is not sufficient to conclude that there is a variation in storminess related to Nortes.

For most of the GoM time series of events related to TCs and their characteristics, the presence of a trend in the data cannot be corroborated by the Mann-Kendall test at the 90 % significance level. However, the basin shows a certain consistency, with no trends (or trends close to zero) for the Coatzacoalcos and Paraiso nodes and increasing trends with increasing distance from these nodes. Overall, with exception of the Bay of Campeche, the results for TC-related events show an increase in wave height in the WCS and the GoM, which, according to Appendini et al. (under review) could be attributed to climate change.

In the WCS, the data confirm the presence of positive trends in the number, SWH, duration and energy of storm events. There is a subtle increase in the number of storms related to TCs, which will result in an increase of one TC-related event every 10 years in Cancun and every 20 years in Tulum. Considering the mean SWH, the obtained trends result in a more evident increase in mean SWH associated with TC events from 1998 and 2008 such as 0.93 m in Tulum and 0.75 m in Cancun.

## Acknowledgments

The Mexican National Council for Science and Technology (CONACYT) and the Universidad Nacional Autónoma de México provided financial support through projects INFR-2014-01-225561 and Proyecto Interno 6602 Dinámica temporal de la vegetación de playas y dunas costeras y su participación como elemento de estabilización en la morfología del frente de playa. The first author is a Cátedras CONACYT researcher under project 1146 Observatorio costero para estudios de resiliencia al cambio climático.

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



**Table 1. Average significant wave height, <SWH>, critical threshold, $H_{threshold}$, and percentage of $H_{threshold}$ exceedance for each node.**

| Location | <SWH> (m) | $H_{threshold}$ (m) | % exceedance $H_{threshold}$ |
|---|---|---|---|
| Matamoros | 1.34 | 2.56 | 4.5 |
| Tampico | 1.06 | 2.10 | 4.5 |
| Veracruz | 0.92 | 2.13 | 4.8 |
| Coatzacoalcos | 0.91 | 2.15 | 4.7 |
| Paraiso | 0.93 | 2.06 | 4.5 |
| Campeche | 1.00 | 2.15 | 4.4 |
| Progreso | 1.15 | 2.38 | 4.3 |
| Holbox | 1.25 | 2.45 | 4.4 |
| Cancun | 1.07 | 1.90 | 3.8 |
| Tulum | 1.05 | 1.87 | 3.9 |

**Table 2. Trends found in the number of Norte events, $SWH_{mean}$ and $SWH_{max}$, $E_{s,mean}$, total $t_s$, $E_{s,max}$ and mean $t_s$ per season. Grey cells indicate that the presence of a trend is confirmed by the Mann-Kendall test at the 90 % confidence level.**

| Node | # Events | $SWH_{mean}$ (m yr$^{-1}$) | $SWH_{max}$ (m yr$^{-1}$) | $E_{s,max}$ (m$^2$ d yr$^{-1}$) | $\sum(t_s)$ (h yr$^{-1}$) | $E_{s,mean}$ (m$^2$ d yr$^{-1}$) | Mean $t_s$ (h yr$^{-1}$) |
|---|---|---|---|---|---|---|---|
| Matamoros | 0.002 | **-0.010** | **-0.033** | -0.343 | -0.38 | -0.125 | -0.05 |
| Tampico | 0.033 | -0.002 | -0.019 | **-0.177** | -0.32 | **-0.065** | -0.11 |
| Veracruz | -0.008 | 0.001 | -0.009 | -0.022 | -0.88 | -0.021 | -0.06 |
| Coatzacoalcos | -0.003 | 0.001 | -0.009 | 0.199 | 0.28 | 0.025 | 0.04 |
| Paraiso | -0.003 | 0.001 | -0.008 | 0.044 | 0.48 | 0.021 | 0.05 |
| Campeche | -0.006 | -0.003 | -0.014 | 0.092 | 2.13 | 0.046 | 0.23 |
| Progreso | 0.010 | -0.005 | **-0.055** | -0.210 | 2.14 | -0.029 | 0.09 |
| Holbox | -0.001 | -0.003 | -0.020 | -0.152 | 1.69 | 0.010 | 0.25 |
| Cancun | 0.001 | 0.006 | 0.013 | 0.019 | -0.06 | 0.036 | -0.24 |
| Tulum | 0.022 | 0.005 | 0.011 | 0.039 | 0.46 | 0.007 | -0.23 |





**Table 3. Trends found in the number of TC events, estimated SWH$_{mean}$ and estimated SWH$_{max}$, estimated E$_{s,mean}$, total t$_s$, estimated E$_{s,max}$ and mean t$_s$ per season. Grey cells indicate that the presence of a trend is confirmed by the Mann-Kendall test at the 90 % confidence level.**

| Node | # Events | $\widehat{SWH}_{mean}(t)$ (m yr$^{-1}$) | $\widehat{SWH}_{max}(t)$ (m yr$^{-1}$) | $\hat{E}_{s,max}(t)$ (m$^2$ d yr$^{-1}$) | $\sum(t_s)$ (h yr$^{-1}$) | $\hat{E}_{s,mean}(t)$ (m$^2$ d yr$^{-1}$) | Mean t$_s$ (h yr$^{-1}$) |
|---|---|---|---|---|---|---|---|
| Matamoros | 0.039 | 0.062 | 0.083 | 0.166 | 0.81 | 0.081 | -0.04 |
| Tampico | 0.044 | 0.051 | 0.073 | 0.209 | 1.30 | 0.172 | 0.43 |
| Veracruz | 0.022 | 0.034 | 0.047 | 0.010 | 0.68 | 0.009 | 0.27 |
| Coatzacoalcos | 0.004 | 0.000 | 0.007 | 0.102 | 0.12 | 0.029 | -0.10 |
| Paraiso | 0.001 | -0.011 | -0.006 | 0.059 | 0.30 | 0.004 | -0.16 |
| Campeche | 0.029 | 0.034 | 0.043 | -0.011 | 0.15 | -0.053 | -0.40 |
| Progreso | 0.046 | 0.034 | 0.059 | 0.480 | 1.82 | 0.185 | 0.13 |
| Holbox | 0.071 | 0.068 | 0.120 | 1.462 | 4.17 | 0.845 | 1.55 |
| Cancun | 0.102 | 0.077 | 0.143 | 0.927 | 4.97 | 0.503 | 2.02 |
| Tulum | 0.054 | 0.075 | 0.127 | 0.642 | 2.68 | 0.490 | 1.45 |

**Table 4. Trends obtained for the time series, excluding years with no events related to TCs.**

| | $\widehat{SWH}_{mean}(t\|storm)$ | $\widehat{SWH}_{max}(t\|storm)$ | $\hat{E}_{s,mean}(t\|storm)$ | $\hat{E}_{s,max}(t\|storm)$ |
|---|---|---|---|---|
| Matamoros | 0.002 | 0.008 | -0.293 | -0.448 |
| Tampico | 0.013 | 0.030 | 0.035 | -0.093 |
| Veracruz | 0.003 | 0.009 | -0.312 | -0.308 |
| Coatzacoalcos | 0.006 | 0.022 | 0.291 | 0.259 |
| Paraiso | 0.001 | 0.022 | 0.528 | 0.350 |
| Campeche | 0.004 | -0.002 | -0.739 | -0.526 |
| Progreso | 0.010 | 0.041 | 0.765 | 0.465 |
| Holbox | 0.026 | 0.078 | 2.422 | 1.589 |
| Cancun | 0.005 | 0.048 | 0.651 | 0.446 |
| Tulum | 0.028 | 0.073 | 0.593 | 0.387 |





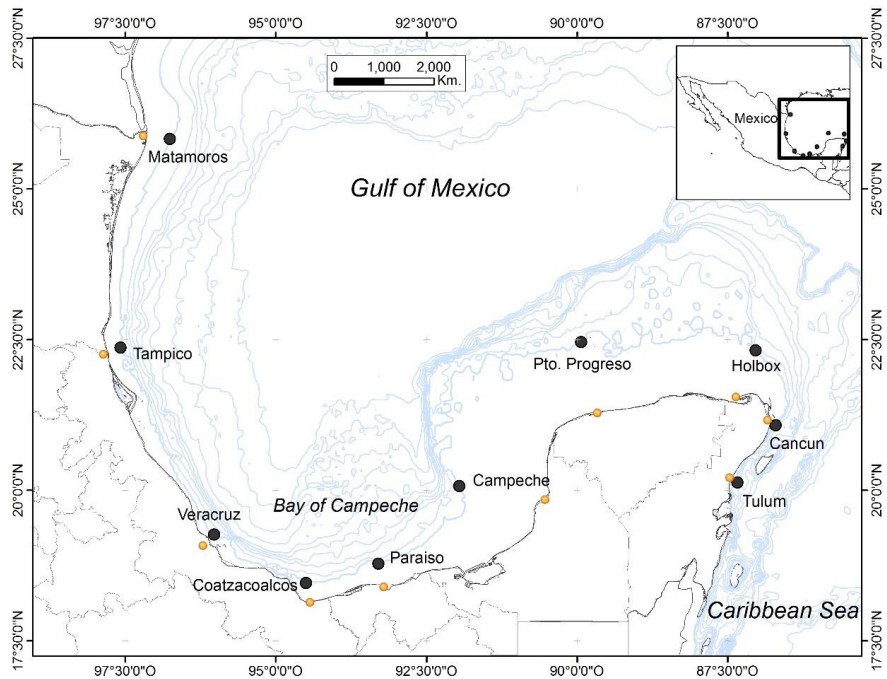

**Fig. 1. Study area with the analysed nodes (black dots) located in the GoM and the Mexican part of the western Caribbean Sea (Cancun and Tulum). Yellow dots indicate the location of nearby settlements.**

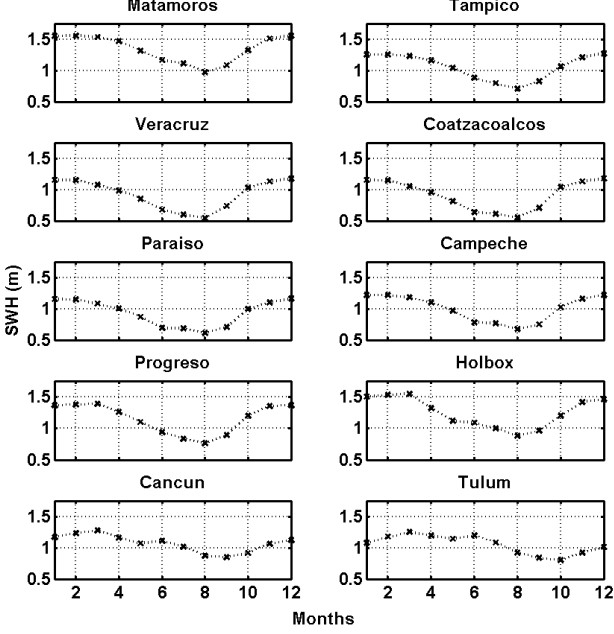

**Fig. 2. Monthly mean SWH for each studied node.**





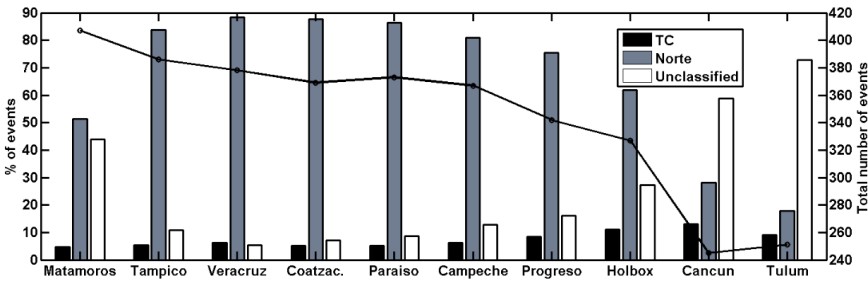

**Fig. 3. Number of extreme wave events per node (solid line) and their percentage of occurrence according to the considered classification (bars).**

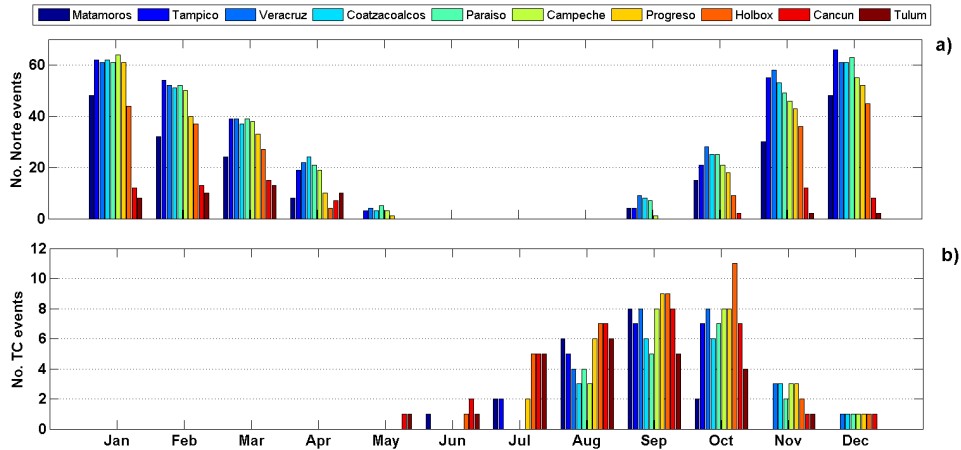

**Fig. 4. Sum of the number of events per month for the entire study period related to (a) Nortes, (b) TCs.**



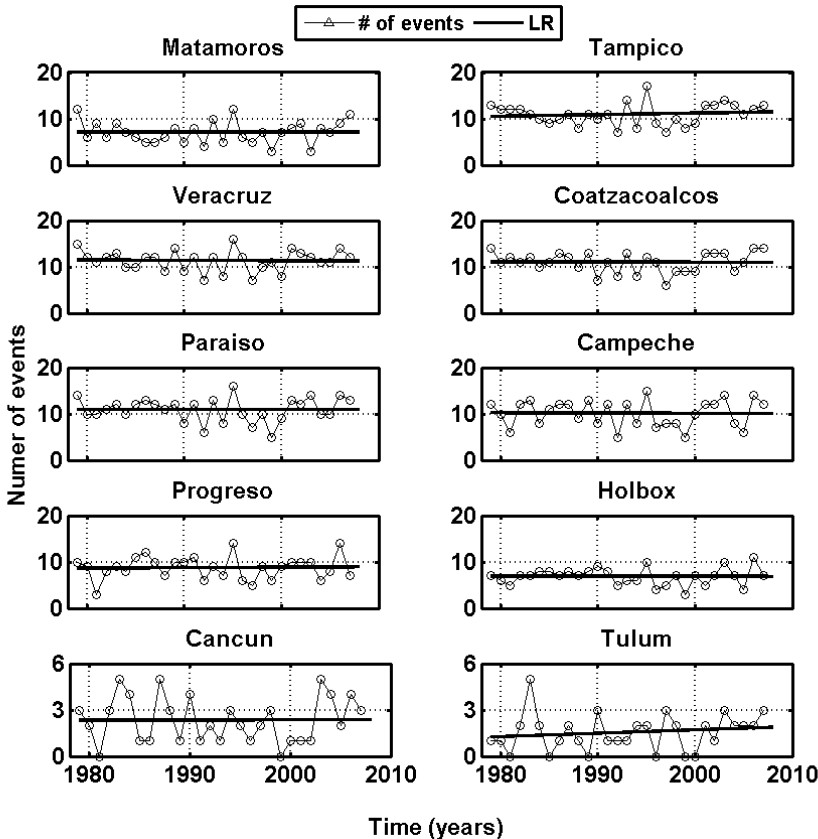

**Fig. 5. Number of events caused by Nortes per season. Simple linear regressions for the data are shown, with the values of the slopes of the regression lines given in Table 2. Notice the different vertical scale used for the Cancun and Tulum nodes.**


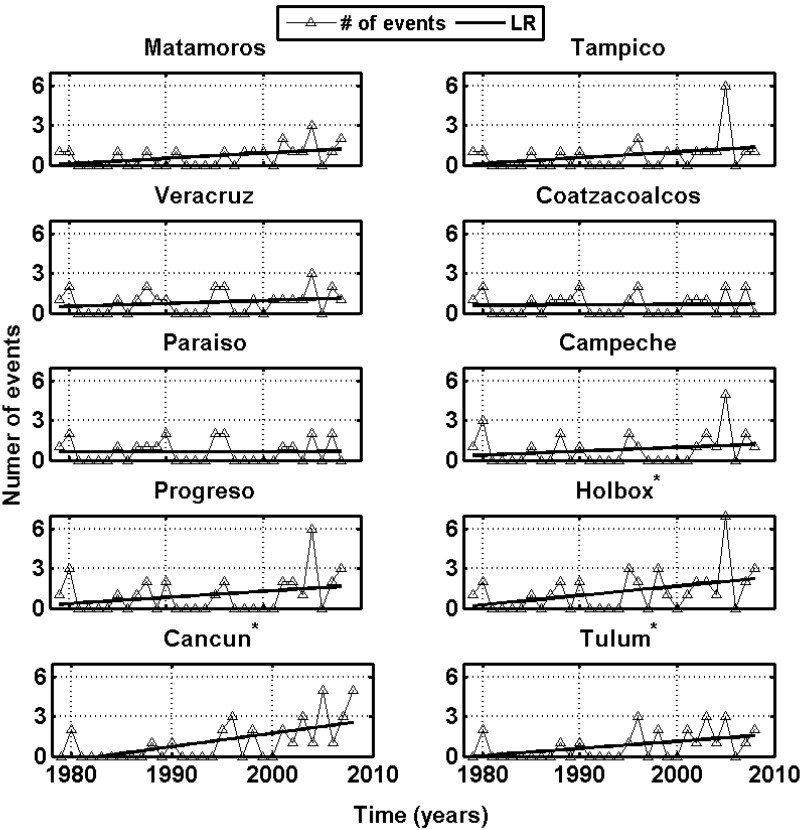

**Fig. 6.** Time series of the number of events caused by TCs each season and trends shown as simple linear regressions. The asterisk indicates the presence of a trend in the time series according to the Mann-Kendall test. The slopes of the regression lines are given in Table 3.

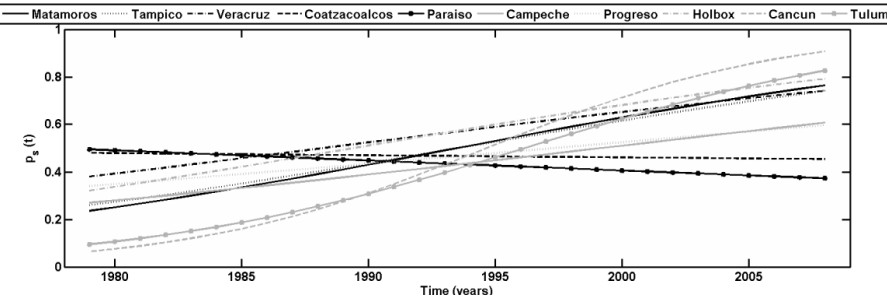

**Fig. 7.** Probability of storm occurrence ($\hat{p}_s(t)$) for the different nodes.





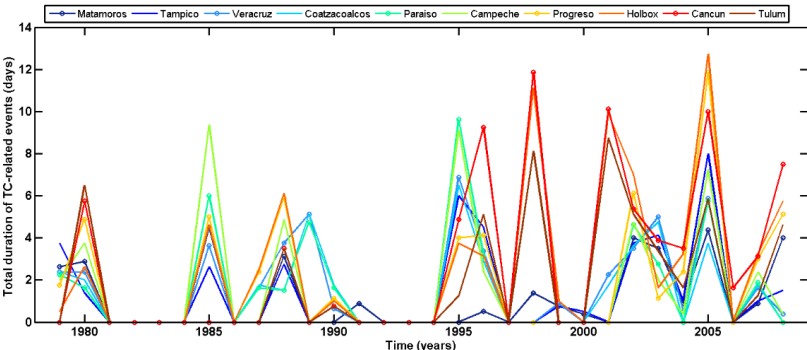

**Fig. 8. Total amount of time (in days) with TC-related storm conditions per season.**

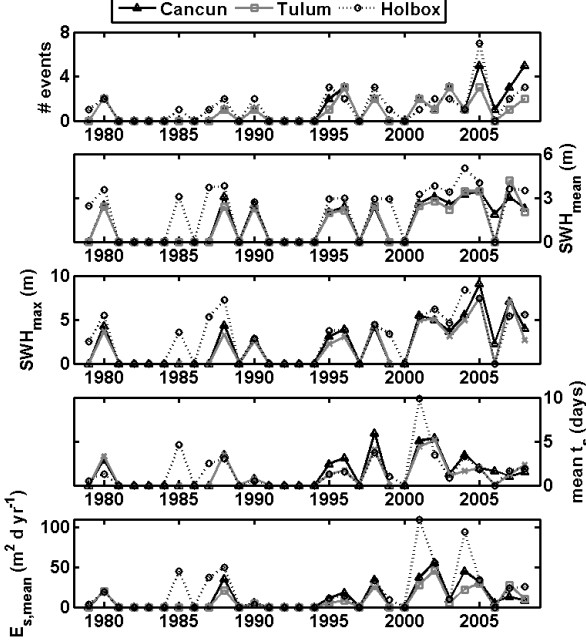

**Fig. 9. Time series of the number of TC events, SWH$_{mean}$ and SWH$_{max}$, mean duration and E$_{s,mean}$ for the Holbox, Cancun and Tulum nodes.**