# Peer review of "Storm-wave trends in Mexican waters of the Gulf of Mexico and Caribbean Sea"

_Natural Hazards and Earth System Sciences, 2016_

## Referee Comment (RC1) · Anonymous Referee #1 · 24 Mar 2017

This manuscripts presents a trend analysis of the nearshore extreme wave climate conditions at Mexican waters based on hindcast data. They focus on the number of events, the mean/maxima wave height and storm energy content, respectively, for two type of events: tropical cyclones and Nortes. A methodology that accounts for the nature of the data is being used and the probability of occurrence involved. I think this study provides relevant results for the study area, where studies as such are scarce and it falls within the Natural Hazards and Earth System Sciences scope. It is also well written and structured. Therefore, I recommend its publication after considering the issues highlighted below. I have three main comments/concerns. The authors use a rather sophisticated method to compute the trend of the storm energy content, involving the probability of occurrence but when showing the results they only show the evolution of the probability of occurrence and a linear-derived trend by calculating the

mean annual rate of increase/decrease using E(t=initial) and E(t=end). I understand the simplification to show the results in a table format but I would suggest to plot the trend evolution of E as directly calculated as well (like in Fig 7 but showing evolution of E). Even if it is qualitatively, I think it is good to know how the trend might deviate from the linear trend, according to this method, especially if extrapolations or qualitative comparisons with future projections are made. A similar methodology could have certainly being used not just for E, but also for # events, since the premises of why using such methodology (rather than a more standard technique) are the same (positive data, many years with zeros). Specially for type of events that are not frequent (ie many years without events), using the linear regressing might lead to meaningless results, for example negative number of events, not just for past extrapolations but also for the period of analysis, as it seems to happen for Cancun (fig 6). An option could be to apply Eq 4 but instead of E(t| storm), having # events (t| storm) In the last paragraph of Section 5.4 it is mentioned the results of Perez et al 2014 regarding the projection under A1B scenario. Then it is roughly compared to the study's results and given the disagreement, it is said that this might be related to the different definition of storm used. This is certainly possible but I believe this is not the only factor (probably not even the most important). Two quite different datasets are compared here: (i) past hindcast (in which greenhouse is not explicitly included) and (ii) a future projection (in which a certain greenhouse scenario is explicitly included). Also the time frames are different. I am not saying this cannot be mentioned but given the existing differences, such discrepancies are reasonable. I would suggest commenting on that by adding other factors that interfere in such comparison. In addition, as a more technical note, I think there is an error at Eq. 3. If E is log-transformed, ie lnE(t|storm) = at + b. Then E(t|storm) =exp(at+b)

---

## Referee Comment (RC2) · Anonymous Referee #2 · 26 Apr 2017

Study focuses on the storm-wave trends in Mexican waters of the Gulf of Mexico and Caribbean Sea. The main outcome is that the number of TC has increased in western Caribbean region, while there are no significant trends in Norte events over the study period. I would recommend to publish this after minor revisions:

Page 1 line 27: Add some more recent studies, if there are any.

Page 2 line 20: Should this reference be Appendini et al. 2014, not 2013.

Page 2 lines 31-31: Correlation to what?

Page 3 lines 15-16: In extreme events even 50 m water depth is not deep, so one will defiantly have shallow water effects.

Page 4 lines 11-12: I really do not understand, what does it mean, that if the SWH is

[Figure]

below threshold for less than 48 hours, two consecutive events are considered as one event. But what happens, when there are 3 such events?

Figure 1: the scale is not correct, must be 100-200 km, not 1000-2000 km.

---

## Referee Comment (RC3) · Anonymous Referee #3 · 26 Apr 2017

Manuscript "Storm-wave trends in Mexican waters of the Gulf of Mexico and Caribean Sea" by Elena Ojeda, Christian Mario Appendini and Ernesto Tonatiuh Mendoza addresses a topical issue of changes in the trends of high wave events in the coastal region. The dataset has been comprehensively analysed and the statistical significance of the trends has been evaluated. The results show that there is increase in the occurrence of high wave events related to TC. The results are of interest to the scientific community and within the scope of nhess. I recommend the paper to be published after minor revision.

Few questions/comments:

Study area, page 2, lines 31-32: Is the accuracy of the hindcast against measurements evaluated in an earlier paper? Does 'best performance' refer to the accuracy of SWH

or also some other parameters and is the evaluation done at all the nodes? Please also give reference.

Hindcast wave data, page 3, line 14-15: Is 50 m depth deep water in the high wave events? What are typical peak periods during these events?

Storm definition, page 4, line 10-11: Is the minimum time between consecutive events, 48 h, based on storm characteristics in GoM? Could you further elaborate the reason behind this selection?

Storm classification, page 4, line 27: Which of the nodes had the lowest/highest number of events occuring during both types of events? Line 33: Could the wind direction criteria be used as main criteria to classify the events?

Conclusion, page 9, line 16: I recommend removing the reference to unpublished work suggesting a link between the presented results and climate change. The paper has enough interesting content even without it.

---

## Author Comment (AC1) · 12 May 2017

The comment was uploaded in the form of a supplement:
http://www.nat-hazards-earth-syst-sci-discuss.net/nhess-2016-392/nhess-2016-392-AC1-supplement.zip

---

## Author Comment (AC2) · 12 May 2017

Dear reviewer,

Thanks for your review and your useful suggestions. The attached files contain a document that addressed each of your questions and suggestions and the new version of the manuscript, which also includes the modifications proposed by two other reviewers

Thanks again, the authors

Please also note the supplement to this comment:
http://www.nat-hazards-earth-syst-sci-discuss.net/nhess-2016-392/nhess-2016-392-AC2-supplement.zip

---

## Author Comment (AC3) · 12 May 2017

Dear reviewer,

Thank you for your suggestions and comments that would improve the final version of this paper. The response to each of your questions and observation are addressed in one of the attached documents, the other contains the manuscript with the suggested modifications and the contributions suggested by two other reviewers.

Best regards, the authors

Please also note the supplement to this comment:
http://www.nat-hazards-earth-syst-sci-discuss.net/nhess-2016-392/nhess-2016-392-AC3-supplement.zip

---

## Author Response (AR1)

Dear reviewers and editor,

First of all, thank you for the suggestions and comments to our manuscript. In this new version the suggestions made by the reviewers have been considered and two major changes to the manuscript have been made: i) we have added a new figure (Fig. 8), which includes the resultant trends obtained for the mean and maxima SWH as well as for the mean and maxima $E_s$ associated to Tropical Cyclone-related events (as recommended by reviewer #1) and, ii) we included the results of analysing the time series of the number of Norte and TC-related storm events using the same methodology than for E and SWH (as suggested by Reviewer #1).

For simplicity, we have answered each of the comments addressed by the three reviewers and the resultant changes are marked in the new version of the manuscript.

REVIEWER #1

**1. The authors use a rather sophisticated method to compute the trend of the storm energy content, involving the probability of occurrence but when showing the results they only show the evolution of the probability of occurrence and a linear-derived trend by calculating the mean annual rate of increase/decrease using E(t=initial) and E(t=end). I understand the simplification to show the results in a table format but I would suggest to plot the trend evolution of E as directly calculated as well (like in Fig 7 but showing evolution of E). Even if it is qualitatively, I think it is good to know how the trend might deviate from the linear trend, according to this method, especially if extrapolations or qualitative comparisons with future projections are made.**

As the reviewer mentioned, the tables were selected as a form to show the results for simplicity. In the new version of the manuscript, a new figure has been added (Fig. 8) which includes the resultant trends obtained for the mean and maxima SWH as well as for the mean and maxima Es associated to Tropical Cyclone- related events.

**2. A similar methodology could have certainly being used not just for E, but also for # events, since the premises of why using such methodology (rather than a more standard technique) are the same (positive data, many years with zeros). Specially for type of events that are not frequent (ie many years without events), using the linear regressing might lead to meaningless results, for example negative number of events, not just for past extrapolations but also for the period of analysis, as it seems to happen for Cancun (fig 6). An option could be to apply Eq 4 but instead of E(t| storm), having # events (t| storm)**

Following this suggestion, the time series of the number of Norte and TC-related storm events were analysed using both, simple linear regression and the same methodology than for E and SWH. The largest changes in the analysis were found on the TC time series, mainly in the Cancun, Holbox, Progreso and Tampico nodes, while Paraiso and Coatzacoalcos showed the minimum changes.
The results are introduced on page 7 (Lines 10-12 and Lines 17-23). Additionally, graphical results have been also included in this version of the manuscript on Fig. 5 and 6 and the trends (estimated in the same manner as for E and SWH) where included in Tables 2 and 3.

**3. In the last paragraph of Section 5.4 it is mentioned the results of Perez et al 2014 regarding the projection under A1B scenario. Then it is roughly compared to the study's results and given the disagreement, it is said that this might be related to the different definition of storm used. This is certainly possible but I believe this is not the only factor (probably not even the most important). Two quite different datasets are compared here: (i) past hindcast (in which greenhouse is not explicitly included) and (ii) a future projection (in which a certain greenhouse scenario is explicitly included). Also the time frames are different. I am not saying this cannot be mentioned but given the existing differences such discrepancies are reasonable. I would suggest commenting on that by adding other factors that interfere in such comparison.**

These differences have been specified at the end of the paragraph (Lines 8-10, page 8).

**4. In addition, as a more technical note, I think there is an error at Eq. 3. If E is log-transformed, ie lnE(t|storm) = at + b. Then E(t|storm) =exp(at+b)**

It certainly was a typo, the term ln was omitted from the first section of the equation but the calculations were correctly made. It has been corrected in the new version of the document.

In this sense, we also changed the nomenclature used in equations 3 and 4 to avoid misunderstandings (E was changed to M so no reference is made to just one of the analysed variables).

REVIEWER #2

**1. Page 1 line 27: Add some more recent studies, if there are any.**

Three new references were added to the introduction related to the contrast between studies of the meteorological effects of storms, where the main interest is focused on the location of the storm strike, and the effect of the wave conditions related to a storm event, that can be observed along distant coasts, well beyond the region of wind stress (Page 1 Line 27).

**2. Page 2 line 20: Should this reference be Appendini et al. 2014, not 2013.**

The reference was corrected (page 2, line 21).

**3. Page 2 lines 31-31: Correlation to what?**

The paragraph has been rewritten (page 2, lines 32-35).

**4. Page 3 lines 15-16: In extreme events even 50 m water depth is not deep, so one will defiantly have shallow water effects.**

Certainly, according to the data a significant percentage of the storm events (Table r2.1) have Tp values associated to transitional waters. This percentage decreases when the Tp values of the entire time series associated to each event are considered. However, the sentence has been omitted from the manuscript (page 3, lines 17-18).

**Table r2.1. Percentage of the peak period associated to TC and Norte events that are not within deep water.**

| Node | % TC | % Norte |
|------|------|---------|
| Matamoros | 94.7 | 57.4 |
| Tampico | 100.0 | 85.1 |
| Veracruz | 87.5 | 81.7 |
| Coatzacoalcos | 84.2 | 88.9 |
| Paraiso | 89.5 | 94.7 |
| Campeche | 73.9 | 90.6 |
| Progreso | 79.3 | 88.8 |
| Holbox | 88.9 | 90.6 |
| Cancun | 9.3 | 0 |
| Tulum | 4.3 | 0 |

**5. Page 4 lines 11-12: I really do not understand, what does it mean, that if the SWH is below threshold for less than 48 hours, two consecutive events are considered as one event. But what happens, when there are 3 such events?**

This sentence needed to be rephrased because it was not clear enough. For consecutive events to be considered independent, the time spam between them must be larger than two days.

We consider that it is more clear in the former version of the manuscript (page 4, lines 14-19): "*Finally, in order to separate consecutive storm events and to assure that the events are statistically independent, an inter-event period of 48 hours was established (Dorsch et al., 2008). This means that consecutive events must be at least 48 hours apart to be considered as independent events, if the SWH is below the threshold for less than 48 hours, consecutive events will be considered as one event associated with a unique meteorological event.*"

**6. Figure 1: the scale is not correct, must be 100-200 km, not 1000-2000 km.**

The scale has been corrected (Fig. 1).

**1. Study area, page 2, lines 31-32: Is the accuracy of the hindcast against measurements evaluated in an earlier paper? Does 'best performance' refer to the accuracy of SWH or also some other parameters and is the evaluation done at all the nodes? Please also give reference.**

Yes, the accuracy of the data was evaluated in a previous paper. The paragraph was not clear enough and it has been rewritten (page 2, lines 32-35).

**2. Hindcast wave data, page 3, line 14-15: Is 50 m depth deep water in the high wave events? What are typical peak periods during these events?**

This was also of concern to Reviewer #2 (question 4) and has been addressed above.

**3. Storm definition, page 4, line 10-11: Is the minimum time between consecutive events, 48 h, based on storm characteristics in GoM? Could you further elaborate the reason behind this selection?**

The definition of storm events varies in the literature with the main parameters being i) the SWH threshold, ii) the minima duration of time during which SWH must remain over the threshold and iii) the minimum time between consecutive storms (e.g., Li, 2011; Mendoza & Jimenez, 2008).
An initial minimum time of 24 hours was selected to differentiate between independent meteorological events but, the more conservative 48 hours criteria was after adopted following previous research such as: Harley et al., (2010): Interannual variability and controls of the Sydney wave climate. *Int. J. Climatol.*; Smits et al., (2005): Trends in storminess over the Nederlands, *Int. J. Climatol.*; or Palutikof, et al., (1999): A review of methods to calculate extreme wind speeds. *Meteorol. Appl.*.
However, the change from 24 to 48 hours did not imply significant changes in our results.

**4. Storm classification, page 4, line 27: Which of the nodes had the lowest/highest number of events occuring during both types of events?**

The lower number of these coincidences occurred in Tulum and Cancun (7 and 12 events, respectively) and the largest numbers in Progreso, Tampico and Veracruz (29, 21 and 21 events, respectively). This information has been included in section 5.2 page 6 lines 1-3 of the new version of the manuscript.

**5. Line 33: Could the wind direction criteria be used as main criteria to classify the events?**
We evaluated the wind direction as an indicator during a certain phase of our research but it was not a good indicator because it included as Norte-related events a significant number of TC-related events.

**6. Conclusion, page 9, line 16: I recommend removing the reference to unpublished work suggesting a link between the presented results and climate change. The paper has enough interesting content even without it.**

The reference has been removed.

I look forward to hear from you,

Sincerely,

Elena Ojeda

[revised manuscript text omitted]